# Electrochemical Sensing of H_2_O_2_ by Employing a Flexible Fe_3_O_4_/Graphene/Carbon Cloth as Working Electrode

**DOI:** 10.3390/ma16072770

**Published:** 2023-03-30

**Authors:** Nebras Sobahi, Mohd Imran, Mohammad Ehtisham Khan, Akbar Mohammad, Md. Mottahir Alam, Taeho Yoon, Ibrahim M. Mehedi, Mohammad A. Hussain, Mohammed J. Abdulaal, Ahmad A. Jiman

**Affiliations:** 1Department of Electrical & Computer Engineering, Faculty of Engineering, King Abdulaziz University, Jeddah 21589, Saudi Arabia; mohammad.mottahir@gmail.com (M.M.A.);; 2Department of Chemical Engineering, College of Engineering, Jazan University, Jazan 45142, Saudi Arabia; 3Department of Chemical Engineering Technology, College of Applied Industrial Technology (CAIT), Jazan University, Jazan 45142, Saudi Arabia; 4School of Chemical Engineering, Yeungnam University, Gyeongsan 38541, Republic of Korea; tyoon@yu.ac.kr; 5Center of Excellence in Intelligent Engineering Systems (CEIES), King Abdulaziz University, Jeddah 21589, Saudi Arabia

**Keywords:** co-precipitation, Fe_3_O_4_/Graphene, flexible carbon fiber cloth, electrochemical, peroxide sensor

## Abstract

We report the synthesis of Fe_3_O_4_/graphene (Fe_3_O_4_/Gr) nanocomposite for highly selective and highly sensitive peroxide sensor application. The nanocomposites were produced by a modified co-precipitation method. Further, structural, chemical, and morphological characterization of the Fe_3_O_4_/Gr was investigated by standard characterization techniques, such as X-ray diffraction (XRD), scanning electron microscopy (SEM), transmission electron microscope (TEM) and high-resolution TEM (HRTEM), Fourier transform infrared (FTIR), and X-ray photoelectron spectroscopy (XPS). The average crystal size of Fe_3_O_4_ nanoparticles was calculated as 14.5 nm. Moreover, nanocomposite (Fe_3_O_4_/Gr) was employed to fabricate the flexible electrode using polymeric carbon fiber cloth or carbon cloth (pCFC or CC) as support. The electrochemical performance of as-fabricated Fe_3_O_4_/Gr/CC was evaluated toward H_2_O_2_ with excellent electrocatalytic activity. It was found that Fe_3_O_4_/Gr/CC-based electrodes show a good linear range, high sensitivity, and a low detection limit for H_2_O_2_ detection. The linear range for the optimized sensor was found to be in the range of 10–110 μM and limit of detection was calculated as 4.79 μM with a sensitivity of 0.037 µA μM^−1^ cm^−2^. The cost-effective materials used in this work as compared to noble metals provide satisfactory results. As well as showing high stability, the proposed biosensor is also highly reproducible.

## 1. Introduction

Nanocomposites (NCs) are used in many applications including adsorption and photodegradation of dyes [1,2,3], pharmaceuticals [4,5], as well as in computers and electronics [6,7], sensors [8,9], heat transfer [10,11], energy storage, renewable energy and supercapacitors [12,13,14]. NCs are also explored in several applications, such as, drug delivery [15,16], hyperthermia [17,18], tissue engineering [19,20], biosensing [21,22,23], energy conversion, and microwave adsorption [24,25]. The NCs are considered as nanomatrix for two or more nanoscale materials which present better performance as compared to their individual counterparts [26]. NCs materials have advantageous and unique properties owing to their boosted physical and chemical properties, high surface area, greater electron transfers ability, and exclusive optical properties [27,28].

Recently, NCs developed using carbon cloth aligned with different nanoparticles for various electrochemical applications. Carbon fiber cloth or/carbon cloth (CFC or/CC) is an alternative to the conventional glassy carbon electrode, screen printed, and carbon paste which are used to fabricate working electrodes in electrochemical-based techniques [29]. The carbon fiber cloth (CFC) is usually made of the polyacrylonitrile-based carbon fiber with high conductivity, excellent flexibility and high porosity with adjustable size proper-ty [29,30]. The CC is employed in various electrochemical-based techniques such as fuel cells, supercapacitors, and sensors and biosensors applications [31,32,33].

Khorablou et al. [30] modified carbon cloth with Au nanoparticles and polythiophene for the detection of methadone. The prepared flexible NCs bear excellent sensitivity and selectivity toward methadone. Similarly, Liu et al. [34] have prepared Cu-coordinated molecularly imprinted polymers (Cu-MIP) and flexible carbon-cloth-supported Ag nanoparticles. The prepared materials were used as working electrodes for detection of perphenazine. The sensor provides satisfactory results with high sensitivity, good selectivity, and promising reproducibility and anti-interference ability. Ahmed et al. [35] proposed a histamine electrochemical biosensor based on a MIP, fabricated using Au inverse opal electrode. To improve the electrical characteristics of substrate, 3,4-ethylenedioxythiophene (EDOT) was electropolymerized on a scaffold of Au electrode. The biosensor exhibited high selectivity, good stability with a limit of detection (LOD) of 1.07 nm, and a linear response from 50 nm–500 µm. Moreover, Xu et al. [36] have proposed an electrochemical enzyme-free biosensor for glucose detection using conductive Ni/Co bimetal metal oxide framework (MOF) which was grown on carbon cloth using facile hydrothermal method. The proposed flexible sensor exhibits a low detection limit of 100 nM (S/N = 3), excellent linearity range of 0.3 μM–2.312 mM, high sensitivity of 3250 μA mM^−1^ cm^−2^, and a fast response time of 2 s under optimized conditions.

Among the most important sensors, the hydrogen peroxide (H_2_O_2_) sensor is in high demand and studied by various researchers. The precise H_2_O_2_ detection is of important concern due to its involvement in many fields, such as, chemical, biological, pharmaceuticals, clinical, environmental, and food processing industry. Therefore, it is necessary to have a H_2_O_2_ sensor with high sensitivity, good reproducibility, fast response, and high stability [37,38]. The sensor based on Fe_3_O_4_ shows excellent stability and electro-catalytic action against a different type of sample [37]. Zhang et al. designed the electrochemical sensor from the nanoparticles of iron oxide-polymer for the detection of H_2_O_2_. Cao et al. employed the co-precipitation method for the preparation of iron oxide NPs and characterized them by several techniques (EDX, SEM, and XRD). Fe_3_O_4_ NPs-based electrode materials showed a good response against the detection of H_2_O_2_. For the detection of H_2_O_2_, high sensitivity and good limit of detection were found [38]. Lin et al. have used the Fe_3_O_4_ and iron oxide/chitosan-based sensor for the detection of H_2_O_2_ in buffer solution. The electro-catalytic performance was measured by cyclic voltammetry at low voltage and also checks the response of H_2_O_2_ with the help of improved glassy carbon (GC) electrode with Fe_3_O_4_. The improved electrode of iron oxide/chitosan revealed good sensitivity and high stability with a considerable limit of detection [39].

In another study, Cao et al. used a co-precipitation method for the fabrication of NCs of ferric oxide-ferrous oxide and analyzed by different techniques (EDX, TEM, SEM, and XRD). The prepared Fe_3_O_4_-Fe_2_O_3_ is used as an electrode for the detection of hydrogen peroxide and finds a LOD of 200 μM in 10 mM buffer solution at pH 7 [40]. Ates et al. fabricated NCs materials from biochar (BC) and Fe_3_O_4_ (Fe) and pasted them on the glassy carbon for the detection of H_2_O_2_. For sensing H_2_O_2_ in the real sample, the detection limit was arranged between 0.5 mM and 10 mM. From the results of reproducibility and sensitivity, it is clearly shown that Fe_3_O_4_-biochar is a good sensing material for the detection of H_2_O_2_ [41]. Cui et al. have also used the co-precipitation method for the fabrication of NCs of cobalt-doped iron oxide and characterized it by several techniques (SEM, XRD, and EDX). The SEM results show the spinal shape NPs forms of Co-doped Fe_3_O_4_ and LOD of H_2_O_2_ were found at 10 mM with a pH of 7 [42]. Sun et al. employed dumbbell-shaped Pt- and Pd-doped iron oxide NPs for the detection of H_2_O_2_ in buffer solution. Different impurities of Pt and Pd in Fe_3_O_4_ were used for H_2_O_2_ sensing and found the LOD of 5 × 10^−9^ M and analyzed that NPs of Pd-Pt-Fe_3_O_4_ are excellent for the detection of H_2_O_2_ in the real sample as well as in the biomedical field [43]. Xu et al. employed the dual mode sensor (colorimetric sensor and electrochemical sensor) for the detection of H_2_O_2_ in the linear range from 1.0 μM to 120 μM. They prepared the NCs of iron oxide-molybdenum disulfide-gold (Fe_3_O_4_-MoS_2_-Au). The limit of detection from electrochemical (EC) and colorimetric sensing was 0.08 μM and 0.109 μM, respectively [44].

Iron oxide NPs, despite their large surface area, tend to aggregate due to large magnetic dipole interaction. Therefore, iron oxide NPs-based sensors show poor sensing performance [45]. Iron oxide nanoparticles (NPs) with carbon-based compounds result in excellent biosensor applications. Due to the unique electrochemical properties of various nano allotropes of carbon reinforced with Fe_3_O_4_ NPs, they show better performance for the detection of H_2_O_2_. Zhao et al. [46] have demonstrated a real sample H_2_O_2_ sensor based on 3D graphene (Gr)-supported Fe_3_O_4_ quantum dots. H_2_O_2_ released from living cells is in very small quantities so it is a huge challenge to detect in situ detection of H_2_O_2_. The Fe_3_O_4_/3DGr NCs was found to be very efficient in situ detection with an outstanding reproducibility and very high sensitivity and selectivity of 274.15 mAM^−1^cm^−2^. The sensor also revealed LOD of ~78 nm and a fast response time of 2.8 s. In a similar study, Yousefinejad et al. [47] have proposed a novel NCs for the detection of H_2_O_2_ in nanomolar range. The proposed NCs comprise carbon dots and iron oxide (Fe_3_O_4_) NPs. The LOD for hydrogen peroxide was determined to be 1.0 × 10^−9^ M and linear range was found to be from 1.0 × 10^−8^ M to1.0 × 10^−3^ M.

Cai et al. have prepared the ternary NCs materials of GO (graphene oxide, Fe_3_O_4_ and PG (pristine graphene) and analyzed by transmission electron microscope (TEM) which clears that the NPs of Fe_3_O_4_ are arranged on the sheets of PG and GO. They employed this sensor for the detection of H_2_O_2_ and dopamine and found the LOD of 90 nM for hydrogen peroxide and 180 nM for dopamine [48]. Zhao et al. have synthesized a sensor by spraying the particles of platinum on NCs of Fe_3_O_4_/rGO and checked the electrochemical performance peroxide sensing. This sensor is very fast and sensitive for the detection of H_2_O_2_ and also very stable and compatible as compared to the other reported sensors. The limit of detection was 1.58 × 10^−6^ M, sensitivity was 6.9 μA mM^−1^ [49]. Cai et al. also made the ternary materials of GO (graphene oxide), PG (pristine graphene), and Fe_3_O_4_ by co-precipitation method and checked by SEM and XPS. With the help of SEM, it concludes that the ternary compound of GO, PE, and Fe_3_O_4_ has outstanding performance for the detection of dopamine as compared to the binary compound of Go and Fe_3_O_4_. The finding limit of detection by this sensor was about 370 nM [50]. Hence, carbonaceous materials provide an excellent platform for biosensors due to their excellent thermal conductivity, high charge mobility and extraordinary electrochemical catalytic activity and shielding properties [51]. However, there is not much literature available on Fe_3_O_4_/Gr NCs for H_2_O_2_ sensing. Therefore, due to the unique chemical structure and electronic properties of NCs materials, it is required to explore such NCs prepared by different methods for H_2_O_2_ sensing.

To further explore the sensing abilities of Fe_3_O_4_/Gr NC, we have successfully synthesized these NCs via chemical co-precipitation method. The electrochemical performance of bare electrodes, Fe_3_O_4_/CC and Fe_3_O_4_/Gr NCs, was analyzed. All sensing parameters such as, LOD, sensitivity, selectivity, stability, and reproducibility were performed. Fe_3_O_4_/Gr/CC shows the better sensing ability such as high selectivity and high reproducibility toward H_2_O_2_. Moreover, Fe_3_O_4_/Gr/CC NCs-based flexible electrodes show a good linear range and a low detection limit for H_2_O_2_ detection.

## 2. Experimental

### 2.1. Materials Used in Synthesis

To obtain Fe^3+^/Fe^2+^ ions we have used 99.9% pure ferric chloride hexahydrate and ferrous sulfate heptahydrate salt which was procured from Sigma-Aldrich (St. Louis, MO, USA). Graphene nanopowder was purchased from Iljin Nano Tech, Seoul, Republic of Korea. HCl (35%) and H_2_O_2_ were purchased from Sinopharm Chemical Reagent Co., Ltd., (Shanghai, China). Polymeric carbon fiber cloth was purchased from AvCarb Material Solutions (Lowell, MA, USA) and polymeric carbon fiber cloth (pCFC or CC) with a thickness of 356 microns and basis weight of 132 g/m^2^, grade of HCB with a plain weave construction.

### 2.2. Synthesis of Nanocomposite

Nanocomposite was synthesized by conventional chemical co-precipitation method. First, in a separate beaker, both salts of iron (Fe^2+^/Fe^3+^) were taken in 1:2 ratios and dissolved in distilled water to obtain a clear aqueous solution. Second, ultrasonic treatment was given to as-received graphene powder in distilled water for 3 h and solid graphene sample was collected from aqueous solution using a centrifuge at 3000× *g* RPM. Both samples of iron oxide were transferred into the aqua treated graphene suspension and the mixtures were stirred for 30 min. A 2M NaOH aqueous solution was added into the mixture drop by drop with continuous stirring until the whole solution turns into black precipitates. Additionally, 10 mL of hydrazine hydrate was added to the solution to stop the oxidation of graphene into graphene oxide during NCs preparation. The precipitates obtained from nanocomposite were washed several times by using distilled water and dried in an oven at 70–80 °C for 4–5 h. A schematic representation of Fe_3_O_4_/Gr NCs fabrication is illustrated in Figure 1.

### 2.3. Characterization

The investigation of nanocomposite was done by X-ray diffraction using Cu Kα radiation (λ = 1.54156 Å) with D8AαS advanced X-ray diffractometer to obtain phase and crystallite size. Particle size and morphology were observed by scanning electron microscopy (FESEM, JEOL, JSM-7600F, Tokyo, Japan). The chemical bonding characteristics were explored using Fourier transform infrared (FTIR) spectroscopy (ATR-FT-IR model Nicolet IS 10). For TEM images, a transmission electron microscope (TEM/HRTEM, JEOL, JEM-2100F) was used and operated at 120 kV for nanocomposite samples. A X-ray photoelectron spectroscopy was carried out by ESCALAB250 equipment outfitted with an Al K X-ray source.

### 2.4. Electrode Fabrication Process

For the fabrication of working electrodes, the surface of the CC is washed with ethanol/water under sonication and dried at room temperature. The 100 mg powder of each material (GO, Gr, and Fe_3_O_4_/Gr) is dispersed individually in the 1 mL of Nafion solution (binder) in order to generate a well dispersed slurry. The prepared slurry was dropped carefully onto the CC (1 cm × 2 cm) with a marked area of 1 cm^2^ and dried at room temperature. The fabricated electrodes were denoted as GO/CC, Gr/CC, and Fe_3_O_4_/Gr/CC and results were compared with bare CC. The fabricated flexible electrodes such as GO/CC, Gr/CC, and Fe_3_O_4_/Gr/CC were used as the working electrodes for the electrochemical measurements. The electrochemical tests were carried out on a potentiostat (VersaSTAT 3, Princeton Research, Princeton, NJ, USA) with a standard three electrode system. The Ag/AgCl was used as a reference electrode (filled with 3.0 M KCl) and a platinum gauge as a counter electrode.

## 3. Results and Discussion

### 3.1. X-ray Diffraction (XRD)

From the XRD patterns, a significant peak of graphene was observed at two theta angles of 26.49° corresponding to the (002) plane of graphene [52], as represented in Figure 1a. The rest of the peaks are related to iron oxide which were observed at 30.19°, 35.55°, 43.18°, 53.63°, 57.11°, 62.82°, and 74.42° corresponding to (220), (311), (400), (422), (511), (440), and (622) crystal planes respectively. The iron oxide peaks intensity and relevant angles match with the JCPDS card no. 01-082-1553 which assures the cubic spinel structure and crystalline nature of the powder nanocomposite sample [53]. The crystal size was calculated using Scherrer equation which was found to be 14.5 nm.

### 3.2. Scanning Electron Microscopy (SEM) and Energy Dispersive X-ray Spectroscopy (EDX)

The morphology of graphene and Fe_3_O_4_/Gr NCs was studied and represented in Figure 1. The Figure 1b shows the SEM image of graphene. The SEM image was taken with a magnification of 15,000 times under an electron beam accelerated with a 20 kV. The image shows multiple graphene sheets spread over each other. At the edge, many aggregates of graphene sheets with multi thick layers are visible. There may be some functional groups found attached on the surface of graphene sheets. It may be due to the fact that during the synthesis process some functional groups such as (−OH and −COOH) still get attached and they can tune the electronic and chemical properties of graphene [54]. The SEM image in Figure 1c shows that at low magnification, several nanosheets of graphene can be seen decorated with Fe_3_O_4_ NPs. The spherical shape of Fe_3_O_4_ NPs in the form of clusters is grafted onto the nanosheets of graphene. Further analysis shows the random distribution of various clusters of Fe_3_O_4_ NPs on the graphene (Figure 1d). The nanospheres of iron oxide are spread on the surface of graphene sheets individually and most of the particles are aggregated. These clusters are helpful during electrochemical reactions by providing a more reactive site and extending the link of the Fe_3_O_4_ nanospheres with the analytes [55]. Additionally, SEM image, FTIR and RAMAN spectra are presented in Figure 2.

The standard method for measuring the elemental make-up and composition of materials in the scanning or transmission electron microscope (SEM/TEM) is energy dispersive X-ray spectrometry (EDXS). EDS is typically used in place of the frequently abbreviated EDXS. An elemental breakdown of the Fe_3_O_4_/Gr nanocomposite is shown in the energy-dispersive X-ray spectra (Figure 1d,e). It thus validates the formation of Fe_3_O_4_/Gr nanocomposite. The elemental composition for Fe, O, and C atoms was obtained and presented in Table 1.

### 3.3. Fourier Transform Infrared Spectroscopy (FTIR)

NCs of iron oxide and graphene were analyzed using FTIR to identify their chemical structure. Figure 2a presents the FTIR spectra of graphene where certain peaks are visible due to the attached functional group. However, peaks at 731 cm^−1^, 1492 cm^−1^, and 1722 cm^−1^ appeared with weak intensity which are attributed to the C=C bending, C-H bending, and C=O stretching, respectively [56,57,58]. The other strong peak is observed at 2345 cm^−1^ which is attributed to the O=C=O and another band appears at 3684 cm^−1^ which is attributed to O-H stretching. However, these bands are due to the partial oxidation of graphene-to-graphene oxide [59]. But most of the peaks which are due to partial oxidation of graphene, vanished in Fe_3_O_4_/Gr nanocomposite. In Figure 2b, the FTIR spectrum displays the peaks of nanocomposites (Fe_3_O_4_/Graphene) at different wavenumbers. Absorption peak of Fe_3_O_4_/graphene nanocomposites is observed at 538 cm^−1^, 1105 cm^−1^, 1592 cm^−1^, 2259 cm^−1^, and 2833 cm^−1^. First peak appeared at 538 cm^−1^ denoted the stretching vibration mode of Fe-O [53,60]. Two broad absorption peaks appear at 2833 cm^−1^ due to the stretching of C-H mode [61] and at 1105 cm^−1^ represents the stretching vibration of C=C mode [53,62] while another peak appear at 1592 cm^−1^ also denoted the stretching vibration mode of C=C [53,63,64]. The peak observed at 2259 cm^−1^ specifying the stretch vibration mode of C≡C [65]. Thus, corresponding peaks are related to the FTIR spectra of Fe_3_O_4_/Gr NCs and differ from the peaks of Gr/GO used as precursor in the synthesis of NCs.

### 3.4. RAMAN Spectroscopy

Figure 2c presents the RAMAN spectroscopy of graphene. RAMAN analysis is a helpful technique which provides eminence features. The peak intensities, positions, and shapes of the curve give useful information in graphene and related materials [66]. From Figure 2c, we observe three different peaks appearing at 1338 cm^−1^, 1571 cm^−1^, and 2688 cm^−1^. The bands appearing at 1338 cm^−1^ and 2688 cm^−1^ are assigned to the D band and 2D band respectively. The band appears at 1571 cm^−1^ is assigned to the G band. The Id/Ig ratio was found to be less than 1 which shows that graphene has less defects [66,67]. The intensity of the 2D band is observed to be less than D and G band which explains that the graphene used in this study is the multilayer graphene (MLG) [66,67].

### 3.5. TEM and HRTEM Analysis

Figure 3 represents the TEM and HRTEM images of the Fe_3_O_4_/Gr matrix which analyzes the particle size and morphology of iron oxide nanoparticles. Figure 3a shows the spherical shaped Fe_3_O_4_ nanoparticles decorated on the graphene sheets ranging from 5–10 nm in size. Figure 3b demonstrates the Fe_3_O_4_ nanoparticles encircled at a point, presented in yellow color while at other points a rectangle is marked which represents the graphene sheets. Figure 3c, d is the enlargement of the point which is marked for Fe_3_O_4_ and graphene sheets respectively. The interplanar lattice space calculated for the Fe_3_O_4_ nanoparticles was found to be 0.25 nm while for graphene sheets it was calculated 0.34 nm.

### 3.6. X-ray Photoelectron Spectroscopy (XPS)

XPS survey scan of Fe_3_O_4_/Gr nanocomposites is shown in Figure 4a. The presence of peaks related to carbon, oxygen, and iron in the prepared material of Fe_3_O_4_/Gr nanocomposites confirms the nanocomposite synthesis. Figure 4b shows that there are 2p core levels of iron, such as, Fe 2p_1/2_ with binding energy of 723.9 eV and Fe 2p_3/2_ with binding energy of 710 eV. In Fe_3_O_4_, Fe exists in two states i.e., Fe^2+^ and Fe^3+^, so the peak observed at 710.2 eV (red color) corresponds to Fe^2+^ states and the peak observed at 712.6 eV (cyan color) corresponds to Fe^3+^ in Fe 2p_3/2_. Moreover, at 718 eV (green color), a satellite peak was observed, indicating that there is partial oxidation of Fe_3_O_4_ to Fe_2_O_3_ [68,69]. Figure 4c, d shows the spectrum of 1s core level of oxygen and carbon, respectively. In Figure 4c the peak of the oxygen atom appears at 530.2 eV (red color) attributed to the Fe-O bond and at 531.2 eV (green color) attributed to O-H bonds. The scan of C 1s core level spectrum is shown in Figure 4d. The peak in C 1s is obtained at 284.7 eV (red color) corresponding to C=C sp^2^ carbon atom and at 285.3 eV (green color) attributed to C-C sp^3^ atom. Moreover, the peak appearing at 287.5 eV (blue color) corresponds to C-O bonds which may be due to the functionalized group attached on graphene [70,71,72]. The XPS data also show that no reaction takes place between graphene nanosheets and iron oxide nanoparticles [11].

## 4. Fabrication of Fe_3_O_4_/Gr /CC and Electrochemical Performances

### 4.1. Fabrication of Fe_3_O_4_/Gr/CC and Sensing toward H_2_O_2_

At room temperature, the fabrication of Fe_3_O_4_/Gr/CC flexible electrode was carried out and the schematic representation is shown in Figure 2 and the detailed process is described in Section 2.4.

On a computer-controlled three-electrode system, the electrochemical sensing experiments of all the fabricated electrodes (GO/CC, Gr/CC, and Fe_3_O_4_/Gr/CC) were performed and results were compared with bare CC. The fabricated electrodes were employed as working electrodes, while a platinum and Ag/AgCl electrode served as the counter and reference electrode, respectively. Using cyclic voltammetry (CV), the testing of each electrode in phosphate buffer solution (PBS) was performed (Figure 5a). The fabricated electrodes (GO/CC, Gr/CC, and Fe_3_O_4_/Gr/CC) were tested in 0.1 M PBS (PH = 7.0) at a scan rate of 0.08 V s^−1^, the electrochemical behaviors of GO/CC, Gr/CC, and Fe_3_O_4_/Gr/CC were examined, and results were compared with CC (Figure 5a). The electro-catalytic properties of the GO/CC, Gr/CC, and Fe_3_O_4_/Gr/CC were examined in the 0.1 M PBS (pH = 7.0) at a scan rate of 0.08 V s^−1^ using cyclic voltammogram (CV). Figure 5a represents the current response of each electrode in PBS and shows pCC has negligible current response. Other electrodes have displayed higher current response. This not only confirms the successful modification of CC’s surface, but also shows the credibility of each electrode for electrochemical performance. This higher electrocatalytic property of Fe_3_O_4_/Gr/CC may be ascribed to the good electrocatalytic properties of the Fe_3_O_4_/Gr. Furthermore, the charge kinetics of the Fe_3_O_4_/Gr/CC, GO/CC, Gr/CC, and CC were explored in 0.1 M PBS (pH = 7.0) using electrochemical impedance spectroscopy (EIS) as shown in Figure 5b. From Figure 5b, we observe that a wide semicircle was observed for GO/CC and Gr/CC (Figure 5b), thus large charge-transfer resistance can be expected. However, CC and Fe_3_O_4_/Gr/CC displayed a low semicircle as compared to GO/CC and Gr/CC, thus would have smaller charge-transfer resistance (Figure 5b). The low-frequency portions of the EIS plot with smaller semicircles of Fe_3_O_4_/Gr/CC have better conducting properties with the diffusion-controlled process and demonstrated better electrocatalytic properties.

Additionally, we looked into the electrochemical capabilities of the Fe_3_O_4_/Gr/CC for sensing of H_2_O_2_ (Figure 5c). Figure 5c displays the CV curves of the Fe_3_O_4_/Gr/CC and CC in 0.1 M PBS with a pH of 7.0 and a scan rate of 0.08 V s^−1^. For the bare CC, a poor current response was observed. For the Fe_3_O_4_/Gr/CC, an improved current response toward the detection of H_2_O_2_ that may arise due to the reduction/oxidation was observed. This high current response originated due to the good electrocatalytic properties of Fe_3_O_4_/Gr/CC. Moreover, we have observed good current response for Gr/CC and CC toward H_2_O_2_. Figure 5d provides the enlargement image of Figure 5c where CV curves for CC/H_2_O_2_, Gr/CC/H_2_O_2_, and Fe_3_O_4_/Gr/CC/H_2_O_2_ are compared.

Using CV, it was also investigated how different H_2_O_2_ concentrations affected the electrochemical performance of Fe_3_O_4_/Gr/CC (Figure 6a). With a scan rate of 0.08 Vs^−1^, we were able to get various CV curves of Fe_3_O_4_/Gr/CC in the presence of varying concentrations of H_2_O_2_ from 10 μM to 110 μM (Figure 6a,b). The observations obviously demonstrated that the current response for reduction has increased at each addition of H_2_O_2_. The calibration plot for linearity check between the peak current response and concentration of H_2_O_2_ revealed that this improved current response was linear (Figure 6c). The linear equation was observed as y = −3.70091E^−5^ x −0.00122, and R^2^ = 0.995.

Additionally, we have also looked at the effects of various scan speeds on the Fe_3_O_4_/Gr/CC toward H_2_O_2_ for the electrocatalytic reduction H_2_O_2_ for sensing application (Figure 7). Figure 7 displays the CVs of Fe_3_O_4_/Gr/CC that were obtained at various applied scan speeds of 20–210 mVs^−1^ in 0.1 M PBS (pH 7.0) containing initial concentration of H_2_O_2_. We noticed that as the scan rate increased from 20 to 210 mVs^−1^, the peak current response at the reduction side increases as the scan speed increases. The obtained results show the responses of peak current at 20 to 210 mVs^−1^ and a linear plot can be obtained. This plot further provides the information that indicates the linear increase in the response of the peak current and suggests the process is diffusion controlled for the sensing of H_2_O_2_ [73].

### 4.2. Limit of Detection and Sensitivity of Flexible Sensor

The limit of detection (LOD) and sensitivity of fabricated Fe_3_O_4_/Gr/CC sensors can be estimated using the previous method reported [74,75]. For this, we used Equations (1) and (2), which were mentioned below, to find out the LOD and sensitivity of the Fe_3_O_4_/Gr/CC for the sensing of H_2_O_2_:Limit of detection (LOD) = 3.3 (σ/S)(1)
Sensitivity = Slope/area of the working electrode(2)
where σ = standard error, and S = slope.

The sensitivity, LOD, and linear range of the Fe_3_O_4_/Gr/CC toward H_2_O_2_ using an electrochemical approach were recorded as 0.037 µA μM^−1^ cm^−2^, 4.79 μM, and 10–110 μM respectively, and the plausible mechanism can be seen in Figure 3. Figure 3 shows the micrographic images which show the transformation of H_2_O_2_ to H_2_O + O_2_ with the electrochemical cyclic voltammetry graph. The graph shows the gradual changes in the detection slope of H_2_O_2_. The reduction/oxidation takes place at the surfaces of an electrode (Fe_3_O_4_/Gr/CC electrode) and helps in the detection of H_2_O_2_ (Figure 3). The reduction of H_2_O_2_ carried out by Fe_3_O_4_/Gr NC via electron transfer leads to the formation of hydroxyl ions (OH^−^), and after combining produced water and oxygen molecules [75].

### 4.3. Test of Analytical Parameters (Selectivity, Repeatability, Reproducibility, and Stability) for Fe_3_O_4_/Gr/CC as Flexible Sensor

One of the most essential analytical parameters for sensors is selectivity. Several bodies, including glucose (Glu), uric acid (UA), ascorbic acid (AA), dopamine (DA), nitrophenol (NP), chlorophenol (CP), nitrophenol (NP), and sodium chloride (NaCl) were applied to test the anti-interference performance of Fe_3_O_4_/Gr/CC, as shown in Figure 8a. There was no noticeable current change after the addition of interfering species, and the current response to the addition of H_2_O_2_ was unaffected, confirming the exceptional selectivity. Further, CV curves were used to test the repeatability of the modified electrode (Fe_3_O_4_/Gr/CC) in order to evaluate the accuracy of the sensor. The results were repeatable, with a relative standard deviation (RSD) of 3.27% (four repeated runs at single electrode) (Figure 8b).

The reproducibility of Fe_3_O_4_/Gr/CC sensor was tested by fabricating six independent electrodes and RSD of current responses for the sensing of H_2_O2 was revealed to be 3.04%. (Figure 8c). A stability was also assessed by evaluating the current response toward H_2_O_2_ for 10 days with the retention of 93.01–94.7% of its initial current response and advising good stability. Thus, the presented sensor has acceptable selectivity, reproducibility, and stability.

In comparison with fabricated electrodes consisting of graphene, CNTs, MOFs, noble metals, and metal oxides, we presented a rather economical sensing platform (Table 2). The CC is a suitable, cost-effective substrate with flexible properties, good conductivity, and tunable size, in contrast to certain typical electrodes [30]. Designing a sensing platform based on the CC with variable sizes results in fast analyte adsorption because of its synergistic and structural features. We can see from Table 2 that low detection limit was obtained using Fe_3_O_4_/Gr/CC for the H_2_O_2_ detection with a good linear range. The noble metal-containing electrodes such as Ag NPs/SnO_2_/GCE, AgNPs/PQ11/graphene, and Pt NPs/UiO-66/GCE were even found comparable in terms of LOD (Table 2). The graphene–MWCNT/GCE, rGO–Fe_2_O_3_–GCE, CuGa_2_O_4_/GCE shows either narrow LOW or comparable as compared to Fe_3_O_4_/Gr/CC. As a result, we have a sensor with good LOD, adaptable characteristics, and a low cost.

## 5. Conclusions

The simplistic and facile fabrication of a new type of flexible Fe_3_O_4_/Gr/CC was achieved by the co-precipitation method. This modified carbon cloth-based electrode exhibited excellent physical and electrochemical properties. When a carbon fiber cloth-based electrode was used as a flexible electrochemical sensor for H_2_O_2_ determination, the modified electrode demonstrated high sensitivity, a wide linear range, and a low detection limit. The results show the high sensitivity of 0.037 µA μM^−1^ cm^−2^ with a limit of detection of 4.79 μM and linearity range was measured from 10 μM to 110 μM. The sensor also revealed good reproducibility and high selectivity. It is expected that the flexible and freestanding Fe_3_O_4_/Gr/CC will provide a modular approach for materials to be manufactured in the future due to its ease of preparation, compatibility with in vivo work, and availability in many potential applications.

## Data Availability

The data presented in this study are available from the corresponding authors upon reasonable request.

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
