# Peer review of "Electrochemical Sensing of H2O2 by Employing a Flexible Fe3O4/Graphene/Carbon Cloth as Working Electrode"

_materials, 2023, doi:10.3390/ma16072770_

Round 1

Reviewer 1 Report

In this manuscript, authors report an electrochemical sensor of Fe3O4 and graphene composite modified carbon cloth electrode for voltammetric detection of H2O2. The content of this manuscript lacks novelty, and is not recommended for publication in “Materials” with the following comments.

1. In Section 2.4 more details about electrode fabrication need to be given, such as the concentration and volume of the slurry.

2. It is suggested that the same characterization methods of Fe3O4/Gr and GR be discussed together, such as SEM and FTIR.

3. The XPS fitting should be performed using professional software with Shirley background subtraction. Besides, there is a small peak at 719 eV in Figure 4b, which is inconsistent with the article "no peak was observed".

4. EIS spectrum is usually obtained in the electrolyte solution of 5 mM [Fe(CN)6]3−/4‑ containing 0.1 M KCl. Why did authors choose PBS in this manuscript?

5. Why amperometry was not used for the qualitative analysis?

6. In Figure 6, the current was measured at different H2O2 concentration. What is the potential selected?

7. The real picture of the working electrode should be provided.

8. Table 1 is not necessary with only one line data and the linear range is not better than the reference.

9. The real sample analysis of this proposed electrode should be performed to elucidate the application.

Author Response

Response to decision letter

Manuscript (materials-2203104)

 Reviewer 1

In this manuscript, authors report an electrochemical sensor of Fe3O4 and graphene composite modified carbon cloth electrode for voltammetric detection of H2O2. The content of this manuscript lacks novelty, and is not recommended for publication in “Materials” with the following comments.

Response: Thank you for your time and efforts to review our submitted manuscript. We are happy to have your rigorous comments/suggestions on our submitted manuscript. We have revised the manuscript as per the comments and provide point-by-point responses.

  1. In Section 2.4 more details about electrode fabrication need to be given, such as the concentration and volume of the slurry.

Response: Thank you, the detailed information about the concentration has been included as per the suggestion in the revised manuscript at page. no. 5.

  1. It is suggested that the same characterization methods of Fe3O4/Gr and GR be discussed together, such as SEM and FTIR.

Response: Thank you for your nice suggestion. We have discussed the same characterizations of Fe3O4/Gr and GR together and rearranged the figures as well. Please refer to the revised version of manuscript.

  1. The XPS fitting should be performed using professional software with Shirley background subtraction. Besides, there is a small peak at 719 eV in Figure 4b, which is inconsistent with the article "no peak was observed".

Response: Thank you for pointing it out. We have done XPS fitting with Shirley background subtraction. From the analysis we observed that a satellite peak existed in the Fe 2P regions, this is due to partial oxidation of Fe3O4 to Fe2O3. We have explained the XPS in the manuscript and all the peaks are discussed. For review, please find the XPS below.

  1. EIS spectrum is usually obtained in the electrolyte solution of 5 mM [Fe(CN)6]3−/4 containing 0.1 M KCl. Why did authors choose PBS in this manuscript?

Response: Thank you for pointing it out. We have tried to investigate H2O2 using 5mM [Fe(CN)6]3−/4 in PBS because of our aim of sensing H2O2 in biological fluids to keep constant pH 7 of the medium and utilizing it for biosensor application.

  1. Why amperometry was not used for the qualitative analysis?

Response: Thank you for your question but due to some dysfunction in the amperometry module of our potentiostat, we are unable to perform an amperometry test at this time.

  1. In Figure 6, the current was measured at different H2O2 concentrations. What is the potential selected?

Response: Thank you. The potential was selected from -1.25 V to 1.25 V.

  1. The real picture of the working electrode should be provided.

Response: Thank you for pointing it out. Please see the image below for fabricated electrodes, as it can be seen the lower portion has a film of Fe3O4/Gr slurry by drop casting technique. Please note that these are two different images of two different electrodes.

  1. Table 1 is not necessary with only one line data and the linear range is not better than the reference.

Response: Thank you for pointing it out. We have deleted table 1 and mention the findings in the text.

  1. The real sample analysis of this proposed electrode should be performed to elucidate the application.

Response: Thank you for pointing it out. Due to some obstacles at present, we are limited to the access of real samples, so right now we are unable to do real sample analysis. We apologized for that.

Reviewer 2 Report

The authors have done novel work to develop a sensor for H2O2 detection. The presentation is neat , however follwoing points need to addressed prior publication

1) Line 64, provide abbreviation for LOD

2) Chemical formula mistakes at sevral place accross the manuscript. Check line 143 to 149 and other place too. 

3) Line 288, Figure 2C and Figure 2D should have been 3C and 3D

4) Figure 5C, the discussion about sensing test results  between  Fe3O4/Gr/CC/H2O2 Fe3O4/Gr/CC was not provided.

5) Figure 5C, the sensor is wrongly notated as F3O4, need to be corrected

6) Table 2 data could have been expanded to include the analyte phosphate buffer solution values and could have been a useful comparison with H2O2. 

7) table 2 column widths need proper presentation

8) Include in abstract the comment that cost effective materials used in this work as compared to noble metals and yet similar results obtained

Author Response

Response to decision letter

Manuscript (materials-2203104)

Reviewer 2

The authors have done novel work to develop a sensor for H2O2 detection. The presentation is neat, however following points need to addressed prior publication

Response: Thank you very much for the comments, suggestions and recommendations for our work to be considered for publication in this journal.

We have carefully addressed the comments/suggestion with point wise response and the manuscript is truly improved after incorporating the changes.

1) Line 64, provide abbreviation for LOD

Response: Thank you for pointing it out, we have included the information accordingly and highlighted.

2) Chemical formula mistakes at several place across the manuscript. Check line 143 to 149 and other place too.

Response: Thank you for pointing it out, we have corrected the subscript from line no. 143 to 149 in the revised manuscript and highlighted.

3) Line 288, Figure 2C and Figure 2D should have been 3C and 3D

Response: Thank you for finding the mistakes. We have corrected the same in line no. 288 in the revised manuscript and highlighted.

4) Figure 5C, the discussion about sensing test results between Fe3O4/Gr/CC/H2O2 Fe3O4/Gr/CC was not provided.

Response: Thank you for pointing it out. The discussion related to the sensing results provided from line no. 354-360 in the revised manuscript. 

5) Figure 5C, the sensor is wrongly notated as F3O4, need to be corrected

Response: Thank you for your notice. We have corrected the figure 5 accordingly.

6) Table 2 data could have been expanded to include the analyte phosphate buffer solution values and could have been a useful comparison with H2O2.

Response: Thank you for your suggestion. Table 2 has been updated as per the suggestion in the revised version of manuscript.

7) table 2 column widths need proper presentation

Response: Thank you for pointing it out. We have corrected table 2 accordingly.

8) Include in abstract the comment that cost effective materials used in this work as compared to noble metals and yet similar results obtained

Response: Thank you for your suggestion. The mentioned text has been included in the abstract.

Reviewer 3 Report

The authors reported an Electrochemical Sensing of H2O2 by Employing a Flexible Fe3O4/Graphene/Carbon Cloth as Working Electrode. This sensor prepared by the nanoprecipitation method shows excellent sensitivity and large detection range. H2O2 is one of the most common reactive oxygen molecules in living organisms and plays an important role in many fields, and the related sensors have great application prospects. However, there are many significant issues in experimental design, article layout, etc. which need to be carefully addressed.

1.      In the characterization section it is recommended to have EDS testing to give the elemental proportions and distribution.

2.      In the sensor sensitivity test, could you add data on the response time of the sensor in the face of continuous changes in H2o2 concentration.

3.      In the selectivity test of the sensor, could you please explain the reason why these interferents were chosen.

4.      The formatting and layout of the icons should be further improved, as there are cases where the title and icons are separated in two pages. (Table 2. Figure 8.)

5.      What are the products of H2O2 in the electrochemical process? The process and mechanism should be further explained.

6.      The flexibility of the electrode in the title is not reflected in the text, is it possible to have the data to illustrate the flexibility.

Author Response

Response to decision letter

Manuscript (materials-2203104)

Reviewer 3

1.      In the characterization section it is recommended to have EDS testing to give the elemental proportions and distribution.

Response: Thank you for your recommendation. We here provide EDS images (see Fig. 1 (e) & (f)) of the prepared nanocomposite with a table of elemental proportions and distribution (See table 1 in the manuscript).

Element

Weight (%)

Atomic (%)

C K

14.98

31.44

O K

26.85

42.30

Fe K

58.17

26.26

Total

99.9

99.9

2.      In the sensor sensitivity test, could you add data on the response time of the sensor in the face of continuous changes in H2o2 concentration.

Response: Thank you for your question but due to abnormal working and some dysfunction in the amperometry module of our potentiostat, we are unable to perform an amperometry test at this time. We apologize for that.

3.      In the selectivity test of the sensor, could you please explain the reason why these interferents were chosen.

Response: We have done interference study to find out the selectivity of the proposed sensor. Since, we have proposed the synthesized materials for biosensor applications, therefore we have chosen some of the important biological analytes such as, AA, UA, DA, Glucose. Although, we have also selected CP and NP as these are organic compounds utilized in medicine and agricultural fields and very commonly studied in electrochemical sensing. 

4.      The formatting and layout of the icons should be further improved, as there are cases where the title and icons are separated in two pages. (Table 2. Figure 8.)

Response: Thank you for pointing it out. The mentioned tables 2 and figure 8 have been revised accordingly.

5.      What are the products of H2O2 in the electrochemical process? The process and mechanism should be further explained.

Response: Thank you for your suggestion. The products in the electrochemical process of H2O2 are H2O and O2. The mechanism of the detection has been explained in the manuscript.

6.      The flexibility of the electrode in the title is not reflected in the text, is it possible to have the data to illustrate the flexibility.

Response: Thank you for your suggestion. Please find out the image below for fabricated electrodes, due to its physical appearance and nature one can visualize the flexibility of the sensor. We have mentioned the flexibility of the film and sensor in the text.

Round 2

Reviewer 1 Report

Although the paper was revised partially, this manuscript is still not recommended for publication in “Materials” in the present version. It is strange that some comments or experimental results cannot be added due to the abnormal working and some dysfunction in the amperometry module of our potentiostat. Please answer these questions with the data provided. Also some new questions are put forward for further improvement.

1. Sections 3.2 and 3.3 should be merged.

2. In Figure 5a, why does GO/CC have a pair of smaller redox peaks in PBS solution?

3. In Figure 5b, EIS plots of CC should be added. Theoretically, GR have excellent electrical conductivity [Biosensors and Bioelectronics 141 (2019) 111384], but the resistance of GR/CC seems to be the largest.

4. The CV curves of CC and Gr/CC in H2O2 should be added in Figure 5c.

5. The current values in Figure 6c was selected according to which potential in fig. 6a.

6. The linear equation should be provided.

Author Response

Response to decision letter

Manuscript (materials-2203104)

Reviewer 1

Although the paper was revised partially, this manuscript is still not recommended for publication in “Materials” in the present version. It is strange that some comments or experimental results cannot be added due to the abnormal working and some dysfunction in the amperometry module of our potentiostat. Please answer these questions with the data provided. Also some new questions are put forward for further improvement.

Response: Thank you for your valuable comments and suggestions. We apologize for the non-functioning of the amperometric module. We have tried our best to provide the data as per the suggestions and now the manuscript is well-improved. Truly, we are thankful to improve the present work.

  1. Sections 3.2 and 3.3 should be merged.

Response: Thank you, the section 3.2 and 3.3 has been merged as per the suggestion.

  1. In Figure 5a, why does GO/CC have a pair of smaller redox peaks in PBS solution?

Response: Thank you for the query.  GO has an oxidation and reduction peak which is the intrinsic feature of its electrochemical properties. In our case, in addition to this, another reduction peak was observed; we were assuming this peak as the impurity peak. Further, we wanted to confirm this behavior. For this, we have performed the test of 3 electrodes in PBS and found the existence of same peak in all cases. So, this assumption was nullified. Thus, we are assume the existence of the peak due to the modification of GO on CC.

  1. In Figure 5b, EIS plots of CC should be added. Theoretically, GR have excellent electrical conductivity [Biosensors and Bioelectronics 141 (2019) 111384], but the resistance of GR/CC seems to be the largest.

Response: Thank you for the suggestion. We have provided the EIS plot of CC in Figure 5b.We agreed GR with the remarks made. We have seen some literature and found that the similar characteristics as reported and which may arise due the formation of defects on the surface during sonication treatment/electrochemical measurement.

(https://doi.org/10.1016/j.pnsc.2015.10.004 https://doi.org/10.1021/acsomega.0c02370).

Figure 5b.EIS plots

  1. The CV curves of CC and Gr/CC in H2O2should be added in Figure 5c.

Response: Thank you for the suggestion. We have added the CC/H2O2 and Gr/CC/H2O2 as per the query in revised figure. In addition, we have provided separately for comparison and more clear observation.

Figure 5. Electrochemical test of fabricated electrodes, (a) Cyclic voltammetry, (b) EIS and (c) sensing test using cyclic voltammetry for fabricated electrodes (CC, Gr/CC, Fe3O4/Gr/CC) in 0.1M PBS (PH=7.0) at a scan rate of 0.08 V s-1.

  1. The current values in Figure 6c was selected according to which potential in fig. 6a.

Response: Thank you for reasonable query. We have used the potential of -0.646V (versus Ag/AgCl) for the peak current plot.

  1. The linear equation should be provided.

Response: Thank you for the suggestion. We have provided the revised figure with highlighting the corresponding values (y=-3.70091E-5 x -0.00122, R2=0.995).

Figure 6c. Calibration plot for varying concentrations versus peak current.
